# Exogenous Sorbitol Application Confers Drought Tolerance to Maize Seedlings through Up-Regulating Antioxidant System and Endogenous Sorbitol Biosynthesis

**DOI:** 10.3390/plants12132456

**Published:** 2023-06-26

**Authors:** Jun Li, Meiai Zhao, Ligong Liu, Xinmei Guo, Yuhe Pei, Chunxiao Wang, Xiyun Song

**Affiliations:** 1College of Agronomy, Qingdao Agricultural University, Qingdao 266109, China; xmguo2009@126.com (X.G.); peiyuhe1980@163.com (Y.P.); 17854266300@163.com (C.W.); 2College of Life Sciences, Qingdao Agricultural University, Qingdao 266109, China; meiai@qau.edu.cn; 3Beijing Vegetable Research Center, Beijing Academy of Agriculture and Forestry Sciences, Beijing 100097, China; liuligong@nercv.org

**Keywords:** maize, sorbitol, polyethylene glycol (PEG), drought tolerance

## Abstract

This study aims to explore the impacts of exogenous sorbitol on maize seedlings under polyethylene glycol (PEG)-simulated drought stress. Six treatments were set: normal condition (CK), PEG (P), 10 mM sorbitol (10S), PEG plus 10 mM sorbitol (10SP), 100 mM sorbitol (100S) and PEG plus 100 mM sorbitol (100SP). Maize seedlings’ growth under PEG-simulated drought stress was significantly inhibited and exogenous sorbitol largely alleviated this growth inhibition. The seedlings under 10SP treatment grew much better than those under P, 100S and 100SP treatments and no significant difference in growth parameters was observed between the control and 10S treatment. The seedlings treated with 10SP had higher contents of soluble sugar, soluble protein, proline, ascorbic acid (AsA), reduced glutathione (GSH), sorbitol and relative water content, higher activities of antioxidant enzymes and aldose reductase, but lower contents of malondialdehyde (MDA), H_2_O_2_ and relative electrical conductivity than those treated with P, 100S and 100SP. qRT-PCR analysis showed that the transcript levels of genes encoding putative aldose reductase (AR) under P treatment were significantly up-regulated in sorbitol-applied treatments. Taken together, the results demonstrated that exogenous sorbitol application conferred drought tolerance to maize seedlings by up-regulating the expression levels of AR-related genes to enhance the accumulation of intracellular osmotic substances such as sorbitol and improve antioxidant systems to tone down the damage caused by drought stress.

## 1. Introduction

Plants always encounter biotic stresses such as drought, salt and extreme temperatures (cold or heat). These stressors negatively impact crop growth, yield and quality [1]. One effective way to settle the problem is to cultivate highly resistant crop varieties, and another is to apply exogenous chemicals directly on the crop. The latter is considered to be a simple strategy in agricultural practices [2]. These applied chemicals are usually the intermediate metabolites of amino acids, sugars and sugar alcohols, including γ-aminobutyric acid (GABA) [2], trehalose and sorbitol [3], ascorbic acid (AsA, antioxidant) [4], 6-Benzylaminopurine (6-BA, phytohormone) [5], methyl jasmonate [6], abscisic acid (ABA, phytohormone) [7], etc. At present, these chemicals’ roles in the enhancement of plant abiotic stress tolerance have been well documented [8,9]. For instance, Tayyab et al. [6] reported that exogenous methyl jasmonate and salicylic acid mitigated drought-induced stress in maize seedlings by modulating the levels of osmolytes and the activities of antioxidant enzymes.

As the main photosynthate of *Rosaceae* fruit trees, sorbitol has been considered to play a dual role in response to environmental stresses in plants. On the one hand, sorbitol serves as an osmotic agent that induces osmotic stress, similar to polyethyleneglycol (PEG). On the other hand, it is a kind of osmoprotectant that enhances the stress tolerance of plants [10,11,12,13,14]. Exogenous sorbitol treatment significantly inhibited seed germination and early seedling growth in maize and *Jatropha curcas* in a dose-dependent manner, and the inhibitory effect was more obvious on shoots than on roots [13,14]. However, several studies exhibited that sorbitol accumulation resulting from salicylic acid (SA) or sufficient potassium treatment improved plant salt or cold tolerance of tomato and *Plantago major* [15,16]. In addition, a certain concentration of sorbitol (10 mM) could increase the salt tolerance of salt-sensitive rice cultivars and the drought resistance of rice-detached leaves, which was realized by up-regulating the antioxidant system of these crops [3,11]. And the transgenic persimmon plants with greatly enhanced sorbitol content exhibited increased salt tolerance but also a dwarfing phenotype [17,18].

Sorbitol is derived from the reduction of glucose in the polyol pathway, which is catalyzed by aldose reductase (AR), with NADPH as a cofactor [19]. In the maize genome, nine genes encode putative aldose reductase (*ZmARs*). Contrary to *AR* genes from other plants, animals and fungi, *ZmAR1* prefers sorbitol to glucose as a substrate [20,21], and further analysis demonstrated that *ZmAR1* might act as a negative regulator in response to salt and drought stresses, which was achieved through reducing sorbitol content in plants [22]. Except for *ZmAR1*, the functions of other AR-related genes in maize are still unclear.

Maize (*Zea mays* L.) is a highly water-consuming crop. In recent years, its production has been frequently threatened by drought stress and this tendency has worsened due to uneven precipitation and global warming [23,24]. Water shortage during the maize sowing stage usually leads to the shortage and uneven growth of seedlings, and even decreases corn yields [25]. Sorbitol can alleviate the drought stress of plants, while little is known so far about the effects on maize germination. Therefore, we applied exogenous sorbitol on maize seedlings under PEG-simulated drought stress to exploit the impacts of sorbitol on maize at the early growing stage. This study aims to provide a new strategy to reduce agricultural and economic losses of maize caused by drought stress in the early stage.

## 2. Results

### 2.1. Exogenous Sorbitol Application Alleviates Growth Inhibition Effect Caused by Drought Stress

PEG-simulated drought stress significantly inhibited maize seedlings’ growth as compared to the control (Figure 1a). The plant length (PL), fresh weight (FW) and dry weight (DW) of roots and shoots under P treatment were reduced by 15%, 46%, 15%, 42%, 26%, and 24%, respectively, compared to control (Figure 1b–d). However, the growth inhibition of maize seedlings caused by drought stress was largely alleviated by the supplement of exogenous sorbitol. Maize seedlings under 10SP treatment grew much better than those under P and 100SP treatments. Compared to P treatment, the PL, FW and DW of root and shoot under 10SP were increased by 17%, 43%, 13%, 53%, 23%, and 21%, respectively. Exogenous sorbitol affected maize seedlings’ growth in a dose-dependent manner. 10S treatment mildly promoted the growth of maize seedlings, but 100S treatment had a strong inhibition effect on maize seedlings’ growth. The maximum growth reduction rate was observed in maize shoots with 100S treatment. The PL, FW and DW of shoots under 100S treatment were reduced by 75%, 34%, and 29%, respectively, over the control.

### 2.2. Effects of Exogenous Sorbitol Application on Chlorophyll Accumulation, Osmolyte Content and Relative Water Content of Maize Seedlings under Drought Stress

Compared to the control, both P and 100S treatments significantly inhibited chlorophyll accumulation, whereas 10S treatment greatly promoted chlorophyll biosynthesis in maize shoots (Table 1). The total chlorophyll contents under P and 100S treatments were reduced by 15% and 31%, respectively, over the control. However, the chlorophyll content under 10S treatment was enhanced, which was an increase of 40% higher than in the control. In PEG-simulated drought conditions, exogenous sorbitol application slightly enhanced the chlorophyll content of maize seedlings, but no significant difference was found between 10SP or 100SP and P treatments. Compared to the P treatment, the total chlorophyll contents under 10SP and 100SP treatments were increased by 8% and 12%, respectively. 

As shown in Figure 2a–c, the osmolyte contents of maize shoots under P, 10S and 100S treatments were significantly enhanced and the maximum increase rate was observed in 10S treatment in comparison to the control. The contents of soluble sugar (SS), soluble protein (SP) and proline under 10S treatment were respectively enhanced by 114%, 39.7% and 243%, over the control. In PEG-simulated conditions, the osmolyte contents under 10SP treatment were much higher than those under 100SP treatment. The SS, SP and proline contents under 10SP treatment were significantly enhanced by 53.5%, 11% and 20.7%, whereas those under 100SP treatment were increased by 34.8%, 1.2% and 13.6%, respectively, over P treatment. The relative water contents (RWCs) under P and 100S treatments were significantly reduced in comparison to the control (Figure 2d), but no difference was observed between 10S and the control. The RWCs under P and 100S treatments were reduced by 36.1% and 52.6%, respectively, over the control. In PEG-simulated drought conditions, the RWC under 10SP treatment was much higher than that under 100SP treatment. Compared to the P treatment, the RWC under 10SP treatment was significantly enhanced by 36.3%, whereas no difference was observed between 100SP and P treatments.

### 2.3. Effects of Exogenous Sorbitol Application on Lipid Peroxidation and Membrane Permeability of Maize Seedlings under Drought Stress

As shown in Figure 3a–c, the contents of H_2_O_2_, malondialdehyde (MDA) and relative electrical conductivity (REC) in maize shoots under P and 100S treatments were significantly enhanced in comparison to the control. The contents of H_2_O_2_, MDA and REC under P and 100S treatments were, respectively, 1.63-, 1.66-, 1.75-, 1.78-, 1.95- and 2.15-times that in the control. 10S treatment caused a mild increase in the contents of H_2_O_2_, MDA and REC when compared to P and 100S treatments. The contents of H_2_O_2_, MDA and REC under 10S treatment were 1.12-, 1.11- and 1.06-times the control, respectively. And no significant difference in REC was observed between the control and 10S treatment. In PEG-simulated drought conditions, exogenous sorbitol application greatly reduced the contents of H_2_O_2_, MDA and REC. The contents of H_2_O_2_, MDA and REC under 10SP treatment were much lower than those under 100SP treatment in comparison to P treatment. The contents of H_2_O_2_, MDA and REC under 10SP treatment were reduced by 13.9%, 26.7% and 35.3%, respectively, over P treatment. 

### 2.4. Effects of Exogenous Sorbitol Application on Antioxidant Systems of Maize Seedlings under Drought Stress

The activities of antioxidant enzymes in maize shoots under P, 10S and 100S treatments were significantly increased and the maximum increase rate was observed in 10S treatment in comparison to the control (Figure 4). The activities of CAT, POD and SOD under 10S treatment were respectively 7.02-, 2.16- and 1.41-times the control. In PEG-simulated drought conditions, exogenous sorbitol application further increased the activities of antioxidant enzymes. The activities of CAT, POD and SOD were much higher in 10SP and 100SP treatments than those in P treatment. For instance, CAT, POD and SOD activities under 10SP treatment were enhanced by 236%, 57%, and 16.9%, respectively, over P treatment. 

The ASA contents of maize seedlings under P, 10S and 100S treatments were significantly enhanced in comparison to the control (Figure 5a). The ASA contents under P, 10S and 100S treatments were respectively 3.62-, 6.16- and 2.59-times that under control. The GSH content and GSH/GSSG ratio under 10S treatment exhibited the greatest increases, which were 516% and 175% higher than in the control (Figure 5b,d). However, the GSSG content under 10S treatment showed the lowest value, which was 43% lower than in the control (Figure 5c). In PEG-simulated drought conditions, exogenous sorbitol application further enhanced the AsA, GSH content and GSH/GSSG ratio but decreased GSSG content. Compared to P treatment, the contents of ASA, GSH and GSH/GSSG ratio under 10SP and 100SP treatments were enhanced by 37%, 25%, 61%, 27%, 7.4% and 26%, respectively. However, the GSSG contents under 10SP and 100SP treatments were reduced by 22% and 15%, respectively, over P treatment.

### 2.5. Effects of Exogenous Sorbitol Application on Sorbitol Content and Aldose Reductase Activities under Drought Stress

The sorbitol content and aldose reductase (AR) activity in maize shoots under P, 10S and 100S treatments were significantly enhanced and the maximum enhancement rate was observed in 10S treatment in comparison to the control (Figure 6). The sorbitol content and AR activity under 10S treatment were respectively 2.8- and 2.17-times that of the control. In PEG-simulated drought conditions, 10SP treatment showed higher sorbitol content and AR activities than 100SP treatment. The sorbitol content and AR activity under 10SP treatment were respectively 1.2- and 1.4-times that under P treatment, whereas those under 100SP treatment were respectively 1.1- and 1.2-times that under P treatment. 

### 2.6. Effects of Exogenous Sorbitol on Transcript Levels of Aldose Reductase (AR)-Related Genes in Maize Seedlings under Drought Stress

Aldose reductase (AR)-related genes in maize shoots exhibited differentially expressed patterns across various treatments (Figure 7). The numbers of significantly up-regulated *ZmARs* genes in P, 10S and 100S treatments were 3, 8 and 5, respectively, compared to control. The transcript levels of *ZmAR2*, *ZmAR3* and *ZmAR5* under P treatment were significantly up-regulated by 151%, 159% and 198%, respectively, over the control. However, the transcript levels of *ZmAR1*, *ZmAR7*, *ZmAR8* and *ZmAR9* under P treatment were greatly down-regulated. *ZmAR4* was slightly up-regulated and *ZmAR6* was mildly down-regulated when exposed to P treatment, but no significant difference in the transcript levels was observed between P treatment and control. Except for *ZmAR9* which was down-regulated, the rest of the eight AR-related genes were significantly up-regulated in 10S treatment. Most AR-related genes under 100S treatment showed similar expression patterns, but the fold changes were much less than those under 10S treatment. In PEG-simulated drought conditions, exogenous sorbitol application significantly increased the transcript levels of all AR-related genes. Compared with P treatment, all *ZmARs* genes were significantly up-regulated under 10SP and 100SP treatment. For instance, the transcript levels of *ZmAR1*, *ZmAR2*, *ZmAR3*, *ZmAR4*, *ZmAR5*, *ZmAR6*, *ZmAR7*, *ZmAR8* and *ZmAR9* under 10SP treatment were respectively 3.2-, 2.4-, 2.3-, 1.7-, 1.3-, 1.4-, 2.7-, 5.3- and 1.5-times that of P treatment. 

## 3. Discussion

Polyethylene glycol (PEG) and sorbitol are two osmotic agents employed to mimic plant drought stress. PEG is commonly used because it does not enter the spaces between cell walls [11,26]. In this study, maize seedlings were severely inhibited by 20% PEG-6000, but the inhibitory effect of PEG was greatly alleviated by the addition of exogenous sorbitol, and 10 mM sorbitol was better than 100 mM sorbitol. Additionally, we find that a higher sorbitol concentration inhibits the growth of maize seedlings, while a lower sorbitol concentration acts as a type of osmotic protector to alleviate drought stress. Based on these findings, we believe that sorbitol affected plant growth in a dose-dependent manner, which aligns with previous results [3,13,14,27].

Chlorophyll is the main photosynthesis pigment in plants. Many studies demonstrate that chlorophyll contents are dramatically reduced when plants are subjected to drought stress [28,29]. Jain et al. [13] reported that the total chlorophyll contents in maize seedlings were greatly decreased by relatively high concentrations of sorbitol. Similar to the results, here we found that chlorophyll contents of maize seedlings were significantly reduced in P and 100S treatments but not in 10S treatment when compared to the control. Some osmolytes, such as soluble sugar, soluble protein and proline have been considered to play important roles in response to osmotic stress, and their contents could be largely enhanced by drought stress [30]. Parallel to the observations, we also found that the contents of soluble sugar, soluble protein and proline were increased by simulated drought stresses (PEG or individual sorbitol treatment). The application of exogenous sorbitol had positively affected the osmolyte contents of maize seedlings subjected to P treatment and a low concentration of exogenous sorbitol was more effective than a high concentration. Relative water content (RWC) is one of the important indicators in determining the plant’s capacity to tolerate water deficiency. Meher et al. [28] reported that the RWC of peanut leaves was significantly reduced in response to an increased concentration of PEG. In this study, we also found that RWC was significantly reduced in P and 100S treatments, but not in 10S treatment. Furthermore, the supplement of 10 mM sorbitol better maintained the water state of maize seedlings subjected to PEG. The possible reason is that the low concentration of sorbitol (10 mM) was beneficial for the accumulation of osmolytes and the maintenance of RWC, thus helping maize seedlings mitigate the damage from drought stress.

Besides osmotic stress, drought generates excess reactive oxygen species (ROS) and then triggers oxidative stress [31,32]. So, in the experiments presented here, the contents of H_2_O_2_, MDA and relative electrical conductivity (REC) were significantly increased in the seedlings treated by PEG or sorbitol only, suggesting that both PEG and sorbitol treatments markedly increased lipid peroxidation. However, one relative study showed that sorbitol did not affect lipid peroxidation in detached rice leaves [11]. In our study, the accumulation of lipid peroxidation in maize seedlings under P treatment was greatly alleviated by adding exogenous sorbitol and a low sorbitol concentration was more efficient than a high concentration. To minimize the damage from ROS, plants have developed two sets of antioxidant systems: one is enzymatic antioxidants such as superoxide dismutase (SOD), peroxidase (POD) and catalase (CAT), and the other is non-enzymatic antioxidants such as AsA and glutathione (GSH) [4,33]. Here, we found that PEG-simulated drought stresses significantly enhanced SOD, POD and CAT activities and AsA and GSH contents. The positive effect of PEG on the antioxidant systems of maize seedlings was strengthened by exogenous sorbitol application and these impacts in 10SP treatment were stronger than those in 100SP treatment. Similar to the results reported by Tayyab et al. [6], exogenous chemical application can enhance plant tolerance capacity to oxidative stresses by up-regulating antioxidant systems.

Considering sorbitol can be partially absorbed by plant cells [11], we measured the sorbitol contents and aldose reductase (AR) activities in maize shoots. Our results showed that PEG-simulated drought stresses greatly increased the sorbitol content and AR activities. This is in line with previous reports that showed that drought stress induced early sorbitol accumulation in loquat leaves [34]. However, the sorbitol content and AR activities did not show an increasing trend with sorbitol concentrations enhancing. This indicates that the application of exogenous 10 mM sorbitol is helpful to the biosynthesis and accumulation of endogenous sorbitol. Furthermore, exogenous sorbitol application significantly enhanced the sorbitol contents and AR activities under P treatment. This is consistent with the previous results that the enhanced sorbitol levels in peach leaves contributed significantly to osmotic adjustment when exposed to drought stress [35]. 

There are nine AR-related genes in the maize genome. Except for *ZmAR1*, the rest of the AR-related genes are not well studied [21,22]. In this study, qRT-PCR analysis demonstrated that the expression pattern of maize AR-related genes varied by treatment. Consistent with the previous results reported by Yang et al. [22], the expression of *ZmAR1* was strongly inhibited by drought stress. The fact that more AR-related genes under 10S treatment than P and 100S treatments were largely up-regulated verifies their roles in enhancing AR activities and endogenous sorbitol content. Furthermore, exogenous sorbitol application up-regulated all AR-related genes’ transcript levels of maize seedlings under PEG-simulated drought stress, suggesting the positive roles of AR-related genes in increasing intracellular osmotic substances such as sorbitol to cope with drought stress. Regarding the roles of these AR-related genes in abiotic tolerance, further efforts are required to validate them in the future.

## 4. Methods

### 4.1. Plant Materials and Drought Treatment

Zhengdan 958 is a popular maize cultivar in China. It was used as plant materials in the study, and the seeds were purchased from Henan Goldoctor Seed Industry Co., Ltd., Zhengzhou, China. 

Before the drought test, the seeds were treated with 0.2% sodium hypochlorite for 10 min for surface sterilization, and then rinsed with sterile distilled water four or five times. Drought treatment was conducted on seed germination papers (38 × 25 cm, Anchor Paper Co., St Paul, MN, USA), which were pre-moistened with 10 mL of water or one of several solutions: (1) distilled water (CK), (2) 20% polyethylene glycol (PEG)-6000 (P), (3) 10 mM sorbitol (10S), (4) 20% PEG-6000 plus 10 mM sorbitol (10SP), (5) 100 mM sorbitol (100S) and (6) 20% PEG-6000 plus 100 mM sorbitol (100SP). Ten sterilized seeds were first rolled in a piece of germination paper, and three rolls with 30 seeds were put into a plastic bag as one treatment. Seeds were cultured in a plant growth chamber at 25 °C under 16 h/8 h of light/dark conditions. Afterwards, seedlings were sprayed with 10 mL of corresponding water or solution every two days per roll. Each treatment had three independent biological replications. 

### 4.2. Measurements of Growth Parameters, Relative Water Content and Sampling

Two weeks later, three seedlings were randomly collected from each treatment and photographed. For the measurements of plant length (PL), fresh weight (FW) and relative water content (RWC), ten seedlings per treatment were selected and separated into shoots and roots. Then, the shoots were hydrated in distilled water for 1 day at 4 °C in darkness to determine the turgid weight (TW). Finally, all samples, including shoots and roots, were oven-dried at 105 °C for 30 min, and then held at 80 °C for 24 h to measure the dry weight (DW). RWC was calculated as RWC= (FW − DW)/(TW − DW). Simultaneously, the left shoots were harvested and immediately frozen in liquid nitrogen for physiological index and molecular measurement. 

### 4.3. Assays of Physiological Indicators

Chlorophyll content was measured according to a previous protocol reported by Arnon [36]. The relative electrical conductivity (REC) was measured by the conductance method. MDA, soluble protein and soluble sugar contents were determined according to the methods described by Yusuf et al. [37], Aminian et al. [38], and Hackmann et al. [39], respectively. Proline content was determined using the acid ninhydrin method described by Benitez et al. [9]. The contents of ascorbic acid (ASA), glutathione (GSH) and GSSG (GSH, oxidized) were determined using the assay kit (Beijing Solarbio Science & Technology Co. Ltd., Beijing, China). H_2_O_2_ content was measured using the methods described by Martinez-Gutierrez et al. [40]. The activities of superoxide dismutase (SOD), peroxidase (POD) and catalase (CAT) were assayed, as described by Usluoglu et al. [41]. The aldose reductase activity and sorbitol content were measured as described by Tari et al. [15].

### 4.4. qRT-PCR Analysis of Aldose Reductase (AR)-Related Genes

Total RNA was extracted using an RNA extraction kit (TaKaRa, Beijing, China) and was reverse-transcribed into cDNA using a PrimeScript RT Reagent Kit (Tiangen, Beijing, China). With *ZmActin* as an internal reference, qRT-PCR was performed to determine the expression levels of AR-related genes using specific primers (Appendix A). These AR-related genes included *ZmAR1* (NM_001112512), *ZmAR2* (NM_001152415), *ZmAR3* (NM_001254921), *ZmAR4* (XM_008676603) and *ZmAR5* (NM_001254922), *ZmAR6* (NM_001112461), *ZmAR7* (NM_001367118), *ZmAR8* (XM_020545520), and *ZmAR9* (NC_050101). The total reaction volume for each qRT-PCR was 20 μL, including 2 μL of diluted cDNA, 0.4 μL of each prime, 0.4 μL passive reference dye, 10 μL ChamQ SYBR qPCR Green Master Mix (Vazyme Biotech Co., Ltd., Nanjing, China) and 6.8 μL double-distilled water. PCR was performed at an Applied Biosystems 7500 Real-Time PCR system and the procedure was done at 95 °C for 10 min, 40 cycles of 95 °C for 10 s, 56 °C for 30 s, and a final step of 72 °C for 30 s. The relative expression level was calculated using the 2^−ΔΔt^ method [42]. 

### 4.5. Statistical Analysis

All the assays were performed in three independent biological repetitions unless otherwise indicated. Data were subjected to the analysis of variance (ANOVA) using SPSS 22.0 software (IBM, Armonk, New York, NY, USA) and means were separated according to the least significant difference (LSD) test at a 0.05 level using the SAS program. 

## 5. Conclusions

As summarized in Figure 8, the increased tolerance of maize seedlings to PEG-induced drought stress was ascribed to the application of exogenous sorbitol. This was because exogenous sorbitol enhanced the contents of non-enzymatic antioxidants and the activities of enzymatic antioxidants, and simultaneously reduced the contents of H_2_O_2_ and MDA in plants, minimizing the oxidative stresses of maize seedlings. Exogenous sorbitol also up-regulated the expression levels of AR-related genes to increase the accumulation of endogenous sorbitol and other osmotic substances such as proline, soluble sugar, soluble protein, etc.

## Figures and Tables

**Figure 1 plants-12-02456-f001:**
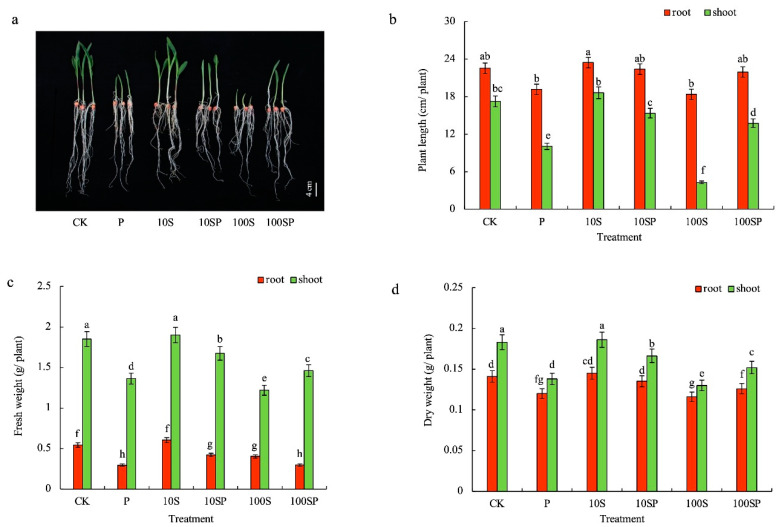
Effects of exogenous sorbitol application on maize seedlings under drought stress. (**a**) Phenotype; (**b**) plant length; (**c**) fresh weight; (**d**) dry weight. Data represent mean ± SD (*n* = 3). Different lowercase letters denote significant differences between treatments according to the LSD test (*p* < 0.05).

**Figure 2 plants-12-02456-f002:**
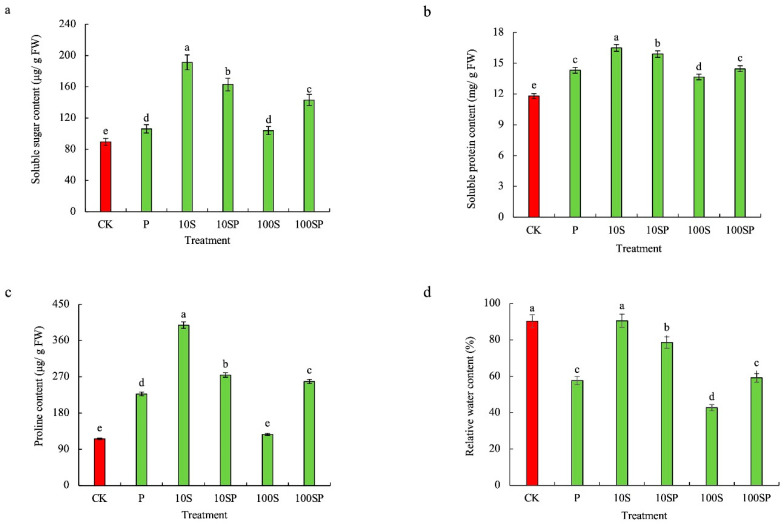
Effects of exogenous sorbitol application on osmolyte and relative water content under drought stress. (**a**) Soluble sugar content; (**b**) soluble protein content; (**c**) proline content; (**d**) relative water content (RWC). Data represent mean ± SD (*n* = 3). Different lowercase letters denote significant differences between treatments according to the LSD test (*p* < 0.05).

**Figure 3 plants-12-02456-f003:**
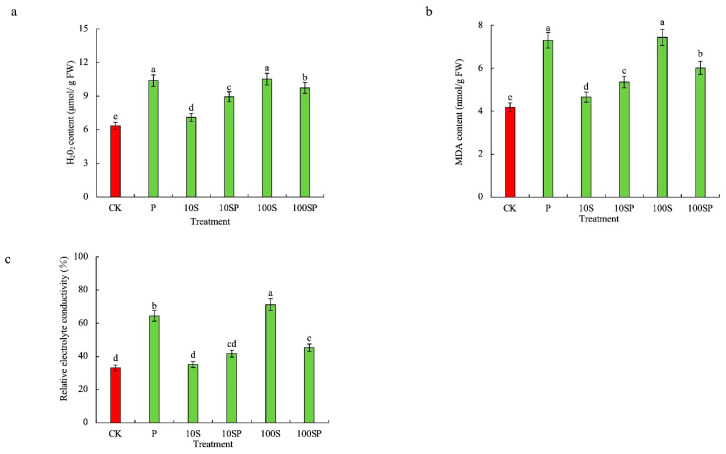
Effects of exogenous sorbitol application on lipid peroxidation and membrane permeability under drought stress. (**a**) H_2_O_2_ content; (**b**) MDA content; (**c**) relative electrical conductivity (REC). Data represent mean ± SD (*n* = 3). Different lowercase letters denote significant differences between treatments according to the LSD test (*p* < 0.05).

**Figure 4 plants-12-02456-f004:**
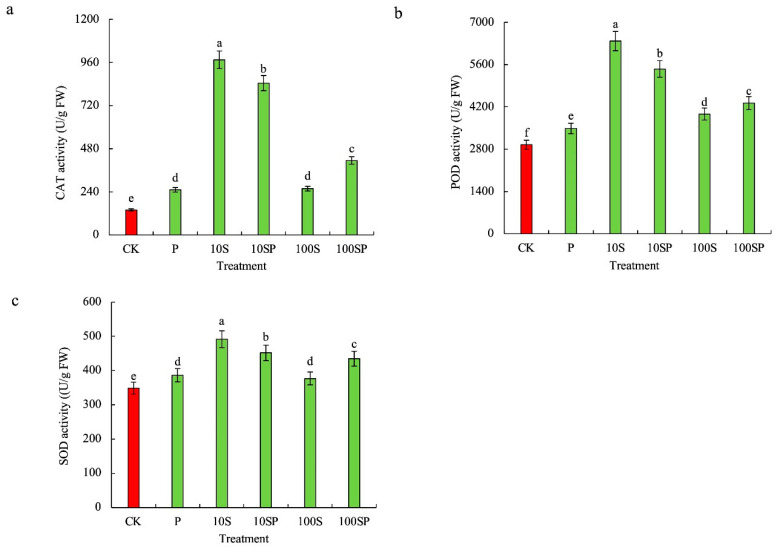
Effects of exogenous sorbitol application on enzymatic antioxidants under drought stress. (**a**) CAT activity; (**b**) POD activity; (**c**) SOD activity. Data represent mean ± SD (*n* = 3). Different lowercase letters denote significant differences between treatments according to the LSD test (*p* < 0.05).

**Figure 5 plants-12-02456-f005:**
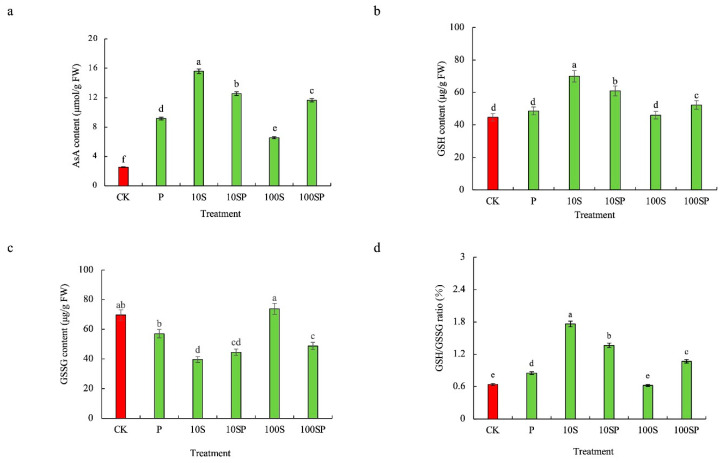
Effects of exogenous sorbitol application on non-enzymatic antioxidants under drought stress. (**a**) AsA content; (**b**) GSH; (**c**) GSSG content; (**d**) GSH/GSSG ratio. Data represent mean ± SD (*n* = 3). Different lowercase letters denote significant differences between treatments according to the LSD test (*p* < 0.05).

**Figure 6 plants-12-02456-f006:**
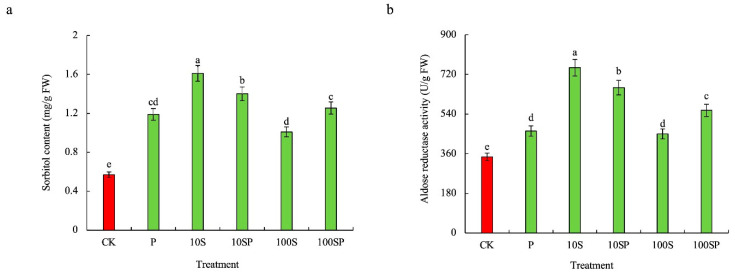
Effects of exogenous sorbitol application on endogenous sorbitol content and aldose reductase (AR) activity under drought stress. (**a**) Sorbitol content; (**b**) aldose reductase (AR) activity. Data represent mean ± SD (*n* = 3). Different lowercase letters denote significant differences between treatments according to the LSD test (*p* < 0.05).

**Figure 7 plants-12-02456-f007:**
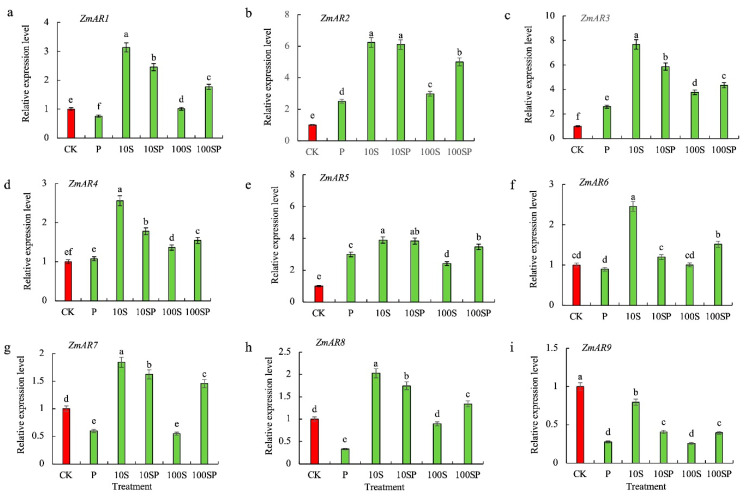
Effects of exogenous sorbitol application on transcription levels of aldose reductase (AR)-related genes. (**a**) *ZmAR1*; (**b**) *ZmAR2*; (**c**) *ZmAR3*; (**d**) *ZmAR4*; (**e**) *ZmAR5*; (**f**) *ZmAR6*; (**g**) *ZmAR7*; (**h**) *ZmAR8*; (**i**) *ZmAR9*. Data represent mean ± SD (*n* = 3). Different lowercase letters denote significant differences between treatments according to the LSD test (*p* < 0.05).

**Figure 8 plants-12-02456-f008:**
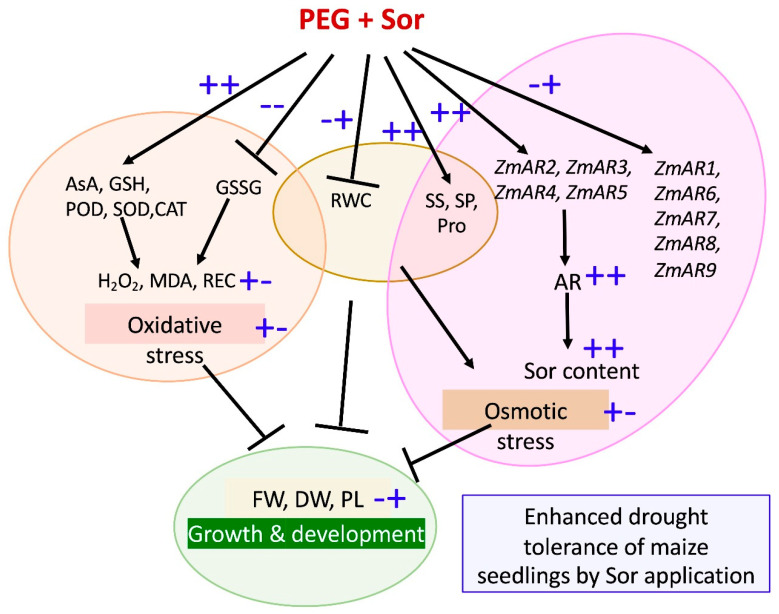
Schematic representation of the mechanisms of enhanced drought tolerance of maize seedlings induced by exogenous sorbitol application. + + or − − indicate the strong increase or decrease treated by both PEG and sorbitol, respectively. + − refer to the initial increase under PEG treatment and subsequent decrease due to the addition of sorbitol. − + refer to the initial decrease under PEG treatment and subsequent increase due to the addition of sorbitol. PEG, Sor, AsA, GSH, GSSG, REC, Pro, AR, POD, SOD, CAT, MDA, RWC, SS, and SP are the abbreviations of polyethylene glycol, sorbitol, ascorbic acid, glutathione, oxidized glutathione, relative electrical conductivity, proline, aldose reductase, peroxidase, dismutase, catalase, malondialdehyde, relative water content, soluble sugar, and soluble protein, respectively.

**Table 1 plants-12-02456-t001:** Effect of exogenous sorbitol application on chlorophyll content in maize shoots under drought stress.

Treatment	Chlorophyll a (mg g^−1^ FW)	Chlorophyll b (mg g^−1^ FW)	Chlorophyll (a + b) (mg g^−1^ FW)
CK	0.98 ± 0.04 ^b^	0.26 ± 0.02 ^b^	1.24 ± 0.03 ^b^
P	0.83 ± 0.05 ^c^	0.23 ± 0.01 ^b^	1.06 ± 0.05 ^c^
10S	1.37 ± 0.05 ^a^	0.37 ± 0.02 ^a^	1.74 ± 0.06 ^a^
10SP	0.90 ± 0.03 ^b^	0.24 ± 0.01 ^b^	1.14 ± 0.05 ^bc^
100S	0.67 ± 0.03 ^d^	0.19 ± 0.02 ^c^	0.86 ± 0.03 ^d^
100SP	0.91 ± 0.04 ^b^	0.27 ± 0.01 ^b^	1.18 ± 0.04 ^bc^

Data represent mean ± SD *(n* = 3). Different lowercase letters denote significant differences between treatments according to the LSD test (*p* < 0.05).

## Data Availability

The datasets generated and/or analyzed during the current study are available from the corresponding author upon reasonable request.

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
