# Peer review of "Exogenous Sorbitol Application Confers Drought Tolerance to Maize Seedlings through Up-Regulating Antioxidant System and Endogenous Sorbitol Biosynthesis"

_plants, 2023, doi:10.3390/plants12132456_

Round 1
Reviewer 1 Report
The manuscript entitled ”Exogenous sorbitol application confers drought tolerance to maize seedlings through up-regulating antioxidant system and endogenous sorbitol biosynthesis” presents data on the effect of sorbitol treatment at two doses on drought tolerance of maize seedlings subjected to simulated water deficit by application of PEG-6000. Increasing drought threat to crop plants forces the need for the search of bioactive compounds most often plant metabolites able to counteract drought stress. Sorbitol is a polyol derived from glucose and serves as osmoregulator, and applied at higher causes osmotic stress itself. The study is well planned and performed, however the manuscript quality is low, mainly due to scanty description and low quality discussion which in parts repeats the results section. Also introduction should present the problem more in depth and concern drought effect of investigated crop and current knowledge on methods to prevent yield losses. Language also needs revision since such phrases as e.g. “no big difference”, “disaster reduction” are not accepted in scientific works. Besides text quality, main issue concern low number of replicates, and at least for biomass assessment the number of replicates should be higher. Secondly, the determination of relative water content should be employed in the study, and chlorophyll content should be divided into a, b, a+b or even a/b to provide more information. The authors should also indicate a potential risk in application of sorbitol which is a development of pathogens using it as a carbon source. Taking into account all mentioned remarks I recommend major revision of the manuscript before reconsideration of the publication in the Plants journal.
Quality of langage low. Needs revission and check by native speaker for a scientific langauge adjustment.
Reviewer 2 Report
review plants-2418757
Exogenous sorbitol application confers drought tolerance to maize seedlings through up-regulating antioxidant system and endogenous sorbitol biosynthesis
General comments
The role of external sorbitol in protecting against stress on the one hand and in reducing the extent of osmotic stress caused by PEG should be emphasized both in the manuscript and in the abstract. A deeper analysis of the relationship between the aldose reductase (AR) genes regulating endogenous sorbitol formation and antioxidants would also be useful in the manuscript. This should be addressed in the discussion section. The marking of differences between the treatments in the figures needs to be reviewed. In many cases, the marking is ambiguous which may affect the conclusion.
Detailed comments
Fig.1b: How should differences be marked: 1) the difference between different treatments in shoot length and root length or 2) between root and shoot length within a given treatment? I suggest that the marking be reviewed and clearly presented.
Also check the marked letters, their font type and size:
fig.1d: dry weight marking of shoot for treatments P and 100S
fig. 2b and c: indicating sugar content and protein content for CK and 100SP treatments
fig. 2d showing the indication of proline content for CK, P, 100S treatments
fig. 4a: the letters at CK; fig. 5c at CK;
fig. 5 d: for CK and 100S treatments
fig. 6a and b: for CK treatment
fig. 7f: for CK and10S treatments
fig.7h: for P and 100SP treatments
in fig. 7d CK notation ef or cf? Check it.
The figure 8 needs to be corrected in the conclusion section: PEG+SOR treatments did not increase the chlorophyll content. Fig. 2 shows that PEG decreased the chlorophyll content but 10SP and 100SP did not change significantly compared to PEG (P) and control (CK) treatments. It must be checked and repaired.
44-45 line: It is true that sorbitol is an osmotic agent but it does not induce osmotic stress rather it has a role in defence. Correct this sentence.
104 line: “sugar content increase by 161% under 100S compared to the control” this is not true. 100S increased the sugar content similarly to P treatment as shown in fig. 2b.
111-112 line: The 1-2% increase in soluble protein content you refer to at 100SP treatment is not considered difference. An increase only applies to 10SP not 100SP. Correct the sentence.
129-131 line: these sentences are contradictory and should be reworded.
168-169 line: GSH/GSSG ratio did not increase in 100S treatment compared to control, see fig. 5d. It needs to be improved.
190-191 line: Check the difference in sorbitol content for P and 100SP treatments. The difference does not seem big (see fig. 6a).
77-88 line: the font type and size appear different from other text.
English style correction required
Round 2
Reviewer 1 Report
The authors greatly improved the manuscript and added requested data. However, the language still needs revision by a native speaker of the field. After language adjustment I recommend to accept the manuscript for publication.
none
